# Machine Learning Improvements to Human Motion Tracking with IMUs

**DOI:** 10.3390/s20216383

**Published:** 2020-11-09

**Authors:** Pedro Manuel Santos Ribeiro, Ana Clara Matos, Pedro Henrique Santos, Jaime S. Cardoso

**Affiliations:** 1Faculty of Engineering, University of Porto, Dr. Roberto Frias Street, 4200-465 Porto, Portugal; jaime.cardoso@inesctec.pt; 2SWORD Health, Sá da Bandeira Street, 4000-226 Porto, Portugal; anaclaramatos@swordhealth.com (A.C.M.); pedro.santos@swordhealth.com (P.H.S.); 3INESC TEC, Dr. Roberto Frias Street, 4200-465 Porto, Portugal

**Keywords:** IMU, human motion tracking, machine learning

## Abstract

Inertial Measurement Units (IMUs) have become a popular solution for tracking human motion. The main problem of using IMU data for deriving the position of different body segments throughout time is related to the accumulation of the errors in the inertial data. The solution to this problem is necessary to improve the use of IMUs for position tracking. In this work, we present several Machine Learning (ML) methods to improve the position tracking of various body segments when performing different movements. Firstly, classifiers were used to identify the periods in which the IMUs were stopped (zero-velocity detection). The models Random Forest, Support Vector Machine (SVM) and neural networks based on Long-Short-Term Memory (LSTM) layers were capable of identifying those periods independently of the motion and body segment with a substantially higher performance than the traditional fixed-threshold zero-velocity detectors. Afterwards, these techniques were combined with ML regression models based on LSTMs capable of estimating the displacement of the sensors during periods of movement. These models did not show significant improvements when compared with the more straightforward double integration of the linear acceleration data with drift removal for translational motion estimate. Finally, we present a model based on LSTMs that combined simultaneously zero-velocity detection with the translational motion of sensors estimate. This model revealed a lower average error for position tracking than the combination of the previously referred methodologies.

## 1. Introduction

Human motion tracking is any procedure that tries to obtain a quantitative or qualitative measure of human movement. It has seen remarkable progress in recent years and is one of the most relevant tasks of motion analysis. Examples of quantitative analysis include the measurement of biomechanical variables such as the angle between joints or parameters that describe the gait of an individual [1,2].

Nowadays, some of the most exciting applications of human motion tracking are medical evaluation, people monitoring, and activity recognition. The continuous recognition of the human activity is fundamental to provide healthcare and assistance services to populations that need to be continuously monitored, such as the elderly or dependent subjects [3]. Furthermore, it can be used in the telerehabilitation field since it allows the rehabilitation of the patient without constant monitoring by a clinician as it can be used as a source of detailed information about the movement of the patient [4].

In recent past, non-visual sensor-based solutions have become a common practice in ambulatory motion analysis since these are portable, can detect motion continuously—even in outdoor environments—and have a low computational cost allowing a real-time feedback [1,5]. With recent advances in microsensor technology, low-power wireless communication and wireless sensor networks, systems using inertial sensors provide an effective and low-cost solution for motion tracking [2].

Inertial Measurement Units based on Micro-Electro-Mechanical Systems (MEMS-based IMUs) have become a popular solution for tracking human motion. IMUs are usually constituted by a set of three accelerometers and three gyroscopes [6]. However, due to the existence of errors such as biases, scale factor imperfections, drifts, misalignments or random noise, these types of sensors suffer from accuracy degradation after extended periods of use [7]. Consequently, the calculation of variables as the velocity, position or orientation through direct integration of the sensor’s signals is an impossible task since these estimations are unreliable after only a few seconds [6]. Thus, currently, there are several efforts to mitigate these errors in the context of motion tracking.

Through the merging of data from three mutually orthogonal gyroscopes and three mutually orthogonal accelerometers of an IMU, it is possible to determine six degrees of freedom (DoF) motion which includes the calculation of both a 3D position and a 3D orientation. The tracking of translational motion is substantially more complicated than the orientation tracking. Since double integration is needed to calculate the displacement of the device, errors rapidly accumulate. A significant problem of this procedure is related to gravity ‘leaking’. A bad estimate of the IMU orientation leads to an incorrect removal of the acceleration caused by gravity. This incorrect removal corrupts the calculation of the velocity and position of the IMU [8,9].

In this paper, we propose the development of Machine Learning (ML) techniques to obtain a better estimate of the 3D human translational motion using inertial data. To achieve these estimates, two different challenges were tackled:A classification problem: The identification of stationary moments can be used in zero-velocity update algorithms allowing a better estimate of position by permitting the correction of the velocity estimates.A regression problem: After identifying the moments in which the IMU is moving, it is essential to understand the translational motion performed by the IMU during those periods. Again, relying solely on the double integration of the acceleration data can lead to wrong estimations of displacements. Using ML techniques, we aim to obtain more robust estimates of displacement.

## 2. Related Work

The application of an inertial navigation system (INS) is the most direct method for estimating the velocity, displacement, and orientation of an IMU using the measurements provided by its accelerometers and gyroscopes. However, the amount of noise present in those inertial measurements, and because the error quickly accumulates, the INS estimates are unreliable after short periods of time [8].

Traditionally, sensor fusion has been used to obtain more accurate estimates. These algorithms include the Kalman Filter, the Complementary Filter, or the Gradient Descent Algorithm [10,11,12]. To improve these sensor fusion methods, more recently, some approaches have been proposed combining constraints from other information sources. For instance, in [13], Solin et al. (2018) combined inertial and video data through the use of an Extended Kalman Filter (EKF). The same authors [9] (2018) used position fixes (a position in time of which the authors are highly confident), loop closures, zero-velocity updates, velocity pseudo-measurement updates, and barometer readings and fused them with the model to improve the estimates for the position, velocity, and orientation performed by the EKF. Inspired by that method, Cortés et al. (2018) [14] proposed a deep learning-based model using a Convolutional Neural Network (CNN) for estimating the velocity of the IMU. This estimated velocity was afterward used as a constraint of an EKF instead of the pseudo-measurements presented by Solin et al. [9], with an increase in performance.

Other approaches fused video and inertial data using neural networks to perform six degrees of freedom (6-DoF) motion estimate. Some examples of this kind of methodologies are presented in [15,16]. However, these examples require a higher computational cost as well as a specific set of conditions for an accurate estimate, such as the fact that the camera needs to be unobstructed.

A type of information that can also be used in sensor fusion algorithms is related to the stationary periods of the IMUs. By determining the periods in which the IMU is still, it is possible to adjust the velocity estimation eliminating the drift. This kind of assumption is also interesting for human motion tracking applications due to the characteristics of the human movement. The identification of the zero-velocity periods is traditionally conducted using threshold-based methods [17,18]. The problem with these algorithms is that they are dependent on the application and IMU performance [19].

Some AI methods have also been proposed for identifying these periods. Park et al. (2016) [20] introduced a methodology for stationary periods identification during human gait using foot-mounted sensors and applying two Support Vector Machines (SVMs)—one was used for activity recognition and the other one was used to classify if the foot is on the ground or not. More recently, Wagstaff and Kelly (2018) [21] proposed a methodology that implements an LSTM (Long-Short-Term Memory Neural Network) to recognize stationary periods in inertial measurements of a foot-mounted IMU and adjusting the velocity with an Extended Kalman Filter.

Other methodologies look at the trajectory estimate problem using end-to-end approaches. These include methods that, using the inertial data, try to estimate both position and orientation of an IMU [8,22,23]. Others try to use the IMU data to guess its velocity and correct the acceleration readings or integrate it to produce a displacement estimation [24,25]. These techniques have consistently obtained better results than simple INS.

Other examples include, for instance, the work developed by Yan et al. [24]: *RIDI* (Robust IMU Double Integration) (2018) is an approach which regresses the velocity vector applying an SVM on inertial data. That velocity is used to correct the accelerometer readings which are integrated twice to provide a displacement estimate. In [26], Chen et al. proposed *IONet* (2018), which uses fixed-sized windows of inertial data to predict the trajectory of the IMU. This way, the input of their neural network is a vector of inertial data, and the output is a polar vector. The authors presented different architectures for the deep neural network: a Recurrent Neural Network (RNN), a Convolutional Neural Network (CNN), a One-Layer Long-Short-Term Memory (LSTM), a Two-Layer LSTM, and a Two-Layer Bidirectional LSTM (BLSTM). The same authors recently presented *L-IONet* (2020) [27]. This is a similar technique, but replaces the used LSTMs with WaveNet [28]. This model allows continuing to take time dependencies into consideration while improving the training time and reducing the computational cost.

Feigl et al. (2019) [25] used a windows-based method involving the signal magnitude vector of the linear acceleration and the angular rate. Several ML-based and deep learning (DL) models were used to regress the signal magnitude vector of human velocity. These include a Classification and Regression Tree (CART), a Support Vector Regressor (SVR), and Gaussian Processes. A DL method was also used applying a CNN combined with a BLSTM.

More recently, Lima et al. (2019) [8] proposed an end-to-end approach that uses data from a low-cost IMU. They used a neural network that combines both CNN and BLSTM layers to estimate the trajectory of the IMU. The loss function of the network was adapted to consider both the translational error and the orientation error. *RoNIN* (2019) was introduced by Yan et al. [29]. Again, inertial data from a smartphone IMU are used to estimate position and orientation. Three different architectures were tested: a 1D version of a Resnet [30], an LSTM, and a Temporal Convolutional Network (TCN) [31]. For the position estimate, the ResNet was trained to predict a translation. LSTM and TCN architectures, on the other hand, were used to regress instantaneous velocities which were integrated. To estimate the IMU orientation, the LSTM architecture was used to regress a 2D vector which represents the sin and the cos values of the heading angle.

Esfahani et al. (2019) [23] presented *AbolDeepIO*, which used DL to estimate the movement of a robot. For that purpose, the authors used several LSTM layers to learn features present in the accelerometer and gyroscopes signals separately. Afterwards, the same authors [22] presented *OriNet* (2019). This framework uses gyroscope readings and a deep learning methodology to improve the orientation quaternion estimate.

All things considered, analyzing the works previously presented, it is possible to conclude that the majority of the works existent in the area are concentrated in tasks that either do not focus on the more complex task of tracking human motion or that—when focused on human motion tracking—lack both movement diversity and data from several body segments.

## 3. Materials and Methods

The proposed solution for 3D human translational motion estimate using inertial data can be divided into two parts. Firstly, a sequence of accelerometer and gyroscope measurements is used to identify the periods in which the IMU is still. In those periods it is assumed that the velocity of the IMU is zero. The next step consisted in using the inertial data from the motion periods to estimate translation. This way, it is possible to estimate the full IMU motion both during zero-velocity periods and motion periods.

### 3.1. Dataset

All experiments were performed using data from the Human Movement and Ergonomics: an Industry-Oriented Dataset for Collaborative Robotics [32]. This dataset was designed to simulate human motion in working conditions in the industry. It contains inertial data collected using a Xsens MVN Link system (Xsens, Enschede, Netherlands) which comprises 17 IMUs placed on several body segments. Various trials were conducted with a total of 13 subjects performing activities such as screwing, untying knots, or carrying loads across a room for a total of 5 h. The participant had to raise their arms, bend their torso, or crouch to perform the mentioned activities. Each trial consisted in performing the mentioned activities in a given sequence, chosen between six predefined sequences. The ground truth was collected with a Qualisys Optical Motion System (Qualisys, Göteborg, Sweden) composed of 43 reflective markers placed in different body parts, the motion of which was recorded using 12 Oqus cameras. Video and data from a prototype e-glove from Emphasis Telematics (Emphasis Telematics, Athens, Greece)—which includes data related to the flexion and pressure of different fingers—are also provided.

It is important to state that, at the beginning of each trial, the system was calibrated, and the Xsens world frame was reinitialized to match the Qualisys world frame. The acceleration due to gravity was already removed from the linear acceleration measurements provided in the dataset, and, in this way, we are dependent of the orientation estimation conducted by the Xsens MVN Link System and subsequent gravity removal process.

The authors provided no information regarding the relative placement of both Xsens IMUs and Qualisys markers. However, analyzing the data provided, it is possible to identify six body placements in which both inertial data and ground truth 3D position are provided: right and left shoulder, right and left upper arm and right and left toe (as represented in Figure 1). Considering these labels, it was considered that the 3D position returned by the Qualisys could be used as ground truth of the position throughout the time of the IMUs in the same placements.

### 3.2. Data Split

To perform the mentioned tasks, only a representative portion of the whole dataset was used due to the high computational cost that using the entire dataset would carry. The generation of the train and test sets is described in Figure 2. The data were split in such a way to guarantee the presence of data from several body segments in both sets. This can increase the generalization ability of the developed models allowing their application to a higher number of body placements. Additionally, this split guarantees that data from a participant are only present in the training set or in the test set. This way, it is possible to ensure that, although the models could learn some particularities of the participant trials present in the training set, those particularities would not increase the test set performance. These particularities can be, for instance, specific errors that can happen during the initialization of the Xsens system that can affect the correct gravity removal or particular placements of the IMUs relative to the Qualisys markers. This split also secures that data from several trials are present in both sets.

### 3.3. Zero-Velocity Detection

The zero-velocity detection problem can be faced as a classification problem in which the goal is to classify a moment into stopped or moving period.

#### 3.3.1. Automatic Labeling of the Qualisys Data

The first step consisted in using the 3D position provided by the Qualisys motion system (ground truth position) to create a binary label—which indicates if that period is a stationary or a motion one. This method was implemented dividing the Qualisys 3D signal into several windows and evaluating the displacement that occurred inside each window. Given a 3D position window of N frames, the label of the window was placed in the frame N/2.

The first step of the algorithm is defining a threshold *thresh*, a “hysteresis term” *T*, and a window size *N*. If the total movement inside the evaluated window is above the specified limit, then it is considered that the respective body segment is moving. Otherwise, it is admitted that the associated body segment is stopped. To introduce hysteresis in the algorithm, it was considered that, if the body segment was stopped, then the movement needed to be *T* times above the defined threshold to change its state.

To evaluate if this algorithm was capable of classifying the Qualisys position data as desired, a randomly selected subset of the dataset was extracted and labeled manually. To perform the manual labeling, the magnitude of the position vector was visually inspected and marked. Out of the 13 subjects who conducted the trials, the manual labeling was performed on one trial for six patients. For the selected trial, the labeling was made in each of the six selected placements (Shoulder (right and left), Upper Arm (right and left), and Toe (right and left)). In total, 36 acquisitions were manually labeled. Finally, the manual labeling was compared with the automatic labeling. The threshold *thresh* and the “hysteresis term” *T* were defined automatically using the ones that maximized the accuracy of the zero-velocity detection.

#### 3.3.2. Fixed-Threshold Zero-Velocity Detectors

To understand the importance of using ML methodologies to segment zero-velocity periods, two different fixed-threshold zero-velocity detectors were used.

##### A: Stance Hypothesis Optimal Estimation (SHOE) Detector

The SHOE [34] detector uses a window of size N of IMU readings to determine if the IMU in experiencing movement or if it is stationary:(1)$$y_{k}=\left\{\begin{array}{c} 1,if\;\frac{1}{N}\sum_{n=k}^{k+N-1}\left(\frac{1}{\sigma_{a}^{2}}{∥a_{n}∥}^{2}+\frac{1}{\sigma_{\omega }^{2}}{∥\omega_{n}∥}^{2}\right)<\gamma \\ 0,otherwise \end{array}\right.$$
in which $a_{n}$ and $\omega_{n}$ represent, respectively, the 3D linear acceleration (with the acceleration caused by gravity already removed) and the 3D angular velocity in the selected window. Each term is divided by the variance of its measurements, $\sigma_{a}^{2}$ and $\sigma_{\omega }^{2}$. If the sum of these terms in the selected window divided by the number of samples N is below the threshold γ, then it is considered that the IMU is stopped, otherwise it is considered that the IMU is experiencing movement [35].

The threshold γ was selected automatically as the one that maximized the zero-velocity detection accuracy in the training set.

##### B: Angular Rate Energy (ARED) Detector

The ARED [34] detector is a binary zero-velocity detector that evaluates the existence of movement using the angular velocity measured by the gyroscope:(2)$$y_{k}=\left\{\begin{array}{c} 1,if\;\frac{1}{N}\sum_{n=k}^{k+N-1}{∥\omega_{n}∥}^{2}<\gamma_{\omega }\\ 0,otherwise \end{array}\right.$$
in which $\omega_{n}$ represents the 3D angular velocity. If the sum of the squared norm of the measurements inside the window of size N is below the threshold $\gamma_{\omega }$, then it is considered that the IMU is stopped, otherwise it is considered that the IMU is experiencing movement [35].

As in the SHOE detector, the threshold, $\gamma_{\omega }$ was selected automatically as the one that maximized the zero-velocity detection accuracy in the training set.

#### 3.3.3. ML Methodologies

##### A: Models

In the next step, different supervised ML classification algorithms were used: a Logistic Regression (LR), a Support Vector Machine (SVM), a Random Forest (RF), a simple LSTM, and a Six-Layer LSTM (6-LSTM). In all of them, as output the binary label which represents if the IMU during that period is experiencing movement or if it stationary was used.

The proposed simple LSTM architecture contained a single LSTM layer with 30 units. This layer included recurrent dropout of 20%, dropping neurons directly in the recurrent connections to reduce overfitting. Finally, a densely-connected layer with only one unit and sigmoid activation was added.

The 6-LSTM model used was the one proposed by Wagstaff and Kelly [21]. It contains six LSTM layers with 80 units per layer. After that, a single densely-connected layer was added to reduce the output to a 2D problem. Considering the way our output was defined (binary output), this last layer is substituted by a densely-connected layer with only one unit and sigmoid activation. Besides, as proposed by the authors, a dropout layer with a rate of 20% was added before the last fully-connected layer to avoid overfitting.

These LSTM-based models were trained using an Adam optimizer [36] and binary cross-entropy loss. Similar to the method implemented by Wagstaff and Kelly [21], a learning rate of $5\times 10^{-3}$ and weight decay of $1\times 10^{-5}$ were used. A batch size of 32 was used in an attempt to balance both the memory requirements and the accurate estimation of the error gradient. Early stopping was implemented to interrupt the training when the validation loss was no longer decreasing over the course of 20 epochs to reduce the risk of overfitting. Finally, gradient clipping was applied in all the methods that used LSTM layers to avoid exploding gradients.

##### B: Features and Cross-Validation

For the LR, SVM, and RF models, several time-domain features were extracted for each time window of each axis signal: mean, maximum, minimum, sum, standard deviation, quartile 0.25, quartile 0.75, median, variance, energy, and maximum discrete difference along the time axis. After extracting these features, pairs of highly correlated features were identified (Pearson’s r > 0.9) in the training set and removed. These same features were also removed from the test set. The label used for each window was the state in the middle of it.

For these three classifiers, a group-three-fold cross-validation was performed to optimize hyperparameters keeping inter-user independence in the validation procedure. The optimization of the hyperparameters of each model was conducted using Bayesian optimization.

For the LR and SVM models, the features present on the training set were standardized by removing the mean and scaling to unit variance. The test set features were standardized using the same parameters. The same schema was used during validation: the validation set in each fold was standardized using the parameters used to standardize the training set in that fold. Since the RF model is tree-based, standardization was not applied.

For the LSTM-based models, the raw signal of each component of the linear acceleration and angular velocity signals were used as input. Again, windows equivalent to the ones used for the automatic labeling of the Qualisys Data were used. The output is likewise the state in the middle of the window.

Due to the computational cost of training neural networks, no group-k-fold was implemented. Instead, for validation purposes, the training set was divided into a training set (composed of seven randomly selected participants) and a validation set (three participants). The six-channel inertial data in the training set were standardized, transforming the values to a distribution with mean 0 and a standard deviation of 1. The same parameters were used to rescale the test set. During validation, the same transformation was applied to the validation set using the parameters used to standardize the train set.

### 3.4. Translational Motion Estimates between Stationary Periods

The outputs of the classification algorithms introduced in the previous section were then used to improve the displacement estimates. This way, it was assumed that when a moment is classified as stopped, then the IMU velocity during that period is zero and no change occurs on its position.

To calculate the changes in position that occurred during the periods of movements two different approaches were used: the first one used a mathematical double integration combined with a linear drift removal and the second one used ML methodologies to introduce a smart double integration.

#### 3.4.1. Double Integration with Linear Drift Removal

As mentioned previously, to calculate the displacements between zero-velocity moments, double integration was performed. Additionally, linear drift removal was executed.

The accumulated drift was calculated as the difference between the velocity estimation at the end of a moving period and at the start of that period. That accumulated drift during each moving period was assumed to be accumulated linearly. That way, the respective drift was removed from each sample. To calculate the corrected position, the drift fixed velocity was integrated.

The velocity correction can be expressed by:(3)$$\mathrm{drift}\;\mathrm{rate}=\frac{v_{e}-v_{b}}{e-b}$$
(4)$$v_{corrected,t_{i}}=v_{t_{i}}-\mathrm{drift}\;\mathrm{rate}\times \left(t_{i}-t_{b}\right),for\;t_{i}=t_{b},...t_{e}$$
in which $v_{e}$ represents the velocity at the end of a movement. $v_{b}$ is the velocity at the beginning of a movement. $t_{b}$ and $t_{e}$ represent the time of the last and first sample of the movement respectively. The corrected velocity for each movement—that occurs between *b* and *e*—is then calculated subtracting the drift rate at the respective time.

#### 3.4.2. ML Methodologies to Perform Double Integration

In this approach, different LSTM-based methods were tested in order to estimate the displacement that took place in a period of movement using as input the inertial data from that period. The main idea behind each model is to assemble a Two-Layer LSTM in which each layer performs an integration that also accounts for the errors present in the linear acceleration signal, and, therefore, can compensate them reaching a better displacement estimate. A representation of this idea is presented in Figure 3.

##### A: Warm-Start with “Integrative Weights”

The first applied method consisted of a Two-Layer LSTM combined with an “integrative initialization”. An overview of the architecture of this approach is represented in Figure 4. This “integrative initialization” means that the neural network weights were initialized in such a way that, without any training, the network would simulate a double integration. Such an initialization makes intuitive sense in the way that the network would start its training already knowing the mathematical operation that it should conduct. This way, during the training procedure, the model would only learn how to account for the errors present in the acceleration data that affect the estimate produced by the double integration operation. The performance of a deep neural model is much reliant on the effective initialization of its parameters, and a smart initialization strategy could improve the performance of the model [37].

This idea is similar to the one existent in transfer learning approaches. In these approaches, the weights from a previous model (that can even be trained on a different domain) are used to initialize the weights from the current model allowing to conduct what is known as “warm start” that corresponds to a smart initialization. In this case, the weights are not from a previous model but the idea of using a smart initialization is the same.

To initialize the weights as desired, the LSTM equations used to determine the candidate cell state and the output were firstly modified. By considering only the example for the first layer, the idea is to use the acceleration as input and to use the cell state to save the velocity in each timestep. In the second layer, the velocity works as input, and the cell state saves the previous position. This way, and assuming that the initial velocity and positions are zero, the output of the first layer is the velocity in each timestep and the output of the second layer is the position in each timestep.

To conduct this analytic weight initialization, the network was simplified. The hyperbolic tangent activation functions of each LSTM block were transformed into linear functions and the weights of the forget, input, and output gates were randomly initialized to close to zero. It is crucial to guarantee symmetry breaking when initializing these weights. If all the weights were initialized to 0 or to the same values, symmetry would exist, and the weight updates during backpropagation might not be effective enough, leading to a poor learning process [38]. This way, the weights of the gates were randomly initialized to different values between $-10^{-6}$ and $10^{-6}$. The biases of these gates were initialized to positive values that would guarantee that all the information would pass through the gates setting the sigmoid function to a value close to 1.

Relatively to the weights and biases related to the cell state, the $w_{c}$ was defined as a matrix containing the sampling period (that multiplies by the current input and is represented by dt) and a value close to zero (that multiplies by the previous output). $b_{c}$ is also initialized to with values between $-10^{-6}$ and $10^{-6}$.

This way, the transformed LSTM equations can be approximated as:(5)$${\tilde{c}}_{t}=\left[0,dt\right]\times {\left[h_{t-1},x_{t}\right]}^{T}+b_{c}=a_{t}\times dt$$
(6)$$c_{t}=f_{t}\times c_{t-1}+i_{t}\times {\tilde{c}}_{t}=1\times v_{t-1}+1\times a_{t}\times dt=v_{t-1}+a_{t}\times dt$$
(7)$$h_{t}=o_{t}\times c_{t}=1\times v_{t-1}+at\times dt=v_{t}$$
in which $a_{t}$ represents the acceleration input of the current timestep and $v_{t-1}$ is the velocity of the previous timestep.

Looking at Equations (Equation 6) and (Equation 7), it is possible to understand that the cell state (**$c_{t}$**) and the output (**$h_{t}$**) both represent the velocity in the current timestep. The same weights were used in the second LSTM layer that is responsible for integrating the velocity into position.

The models were trained using the accelerations of a moving period in each axis as input and the displacement in the respective axis as output. This way, the acceleration of a certain moving period on the y-axis was used as input, and the displacement on the same period on the y-axis was used as output. Information from all the axes was used to train a single model.

After the Two-Layer LSTM, a densely connected layer with no activation was added to reduce the dimensionality of the output to 1D. Again, the weights of this dense layer were initialized, maintaining the purpose of the warm start of the previous layer simulating a double mathematical integration.

##### B: Warm-Start with “Pre-Trained Weights”

The second approach was very similar to the previous one, as described in Figure 5. The goal was also to perform a warm-start to a Two-Layer LSTM model in such a manner that the network, without any training, would resemble a double mathematical integration.

However, in this approach, no modifications were introduced in the LSTM equations, and the weights initialization was not performed manually. This allows maintaining the non-linear activation functions that in the previous approach were replaced by linear functions (Equations (Equation 6) and (Equation 7)).

In the present case, two different ML models were trained: firstly, a One-Layer LSTM followed by a densely connected layer with no activation was trained to learn how to perform a simple integration. This means that the input of the model was the acceleration in a single axis, and its output was the velocity estimated by simple integration. Inertial data from the three-axis was used to train this model. This way, it was expected that this first model would learn how to perform a simple mathematical integration.

Afterwards, the second ML model, a Two-Layer LSTM model (with each LSTM layer followed by a densely connected layer with no activation) was initialized with a warm start using the previously trained weights. This means that each LSTM layer weights were initialized with the final weights of the firstly trained One-Layer LSTM model. Again, it was expected that this warm start would provide a starting point to the neural network in such a manner that the network would start its training already knowing how to perform a double mathematical integration.

Once more, the model was trained using accelerations of a moving period in each axis as input and the displacement in the same axis as output. Information from all the axes was used in the same model.

Another version of this approach was also tested. It included the training of an equal model (One-Layer LSTM + densely connected layer with no activation) with the single-axis linear acceleration as input. However, instead of the velocity calculated through a simple integration, the output used for the first model was the drift corrected velocity (calculated with simple integration and corrected using the drift correction described in Section 3.4.1). The weights from this model were then used to initialize the first layer (LSTM + densely connected layer) of the second ML model. In theory, this should provide the neural network a starting point at which it already knows how to perform an integration from acceleration to velocity with the benefit of removing the drift. The weights used in the second layer of this version were the weights used in both layers of the previous version. The “drift removed weights” were not used in the second layer since that particular correction (drift removal) is specific of the integration from acceleration to velocity.

##### C: No Customized Initialization

The third approach used no warm-start. The neural network structure is the same as the used in the previously presented methodologies: a Two-Layer LSTM followed by a densely connected layer, as represented in Figure 6.

However, in this approach, different inputs and outputs from the ones used in the previous approaches were experimented. The inputs utilized were different combinations of the 3D linear accelerations, 3D angular velocity, and 3D position estimated by double integration. This means that three different models were trained: the first one using the linear acceleration in the three axes as input, the second using the 3D linear acceleration and the 3D angular velocity as input, and the third one using the 3D linear acceleration, the 3D angular velocity, and the 3D estimated position (through double integration) as inputs. All these models were trained to predict the 3D displacement.

##### D: Common Experimental Methodology

All the above-explained models were trained using an Adam optimizer [36] and mean squared error (MSE) loss. As with the methods used in the previous section, a learning rate of $5\times 10^{-3}$, and weight decay of $1\times 10^{-5}$ were applied. In these experiments, a batch size of 1 was used due to the different sized inputs.

The same split as described in Figure 2 was applied. The training set was then split with the data from seven patients for training and the data from the other three patients for validation. Early stopping was also carried out to interrupt the training when the validation loss was no longer decreasing throughout 20 epochs to reduce the risk of overfitting. Finally, gradient clipping was applied to avoid exploding gradients.

The selection of the moving periods was conducted using the reference from the classification models. This data selection means that only the data from moving periods determined by the ground truth was used to train and evaluate the models mentioned above. Consequently, the output was the displacement that occurred during that period. This way, imagining a moving period of 200 samples, the input, depending on the model, is composed of the inertial data during that period. The output also contains 200 samples. In the output, each timestep represents the displacement that happened since the beginning of that moving period.

### 3.5. Simultaneous Zero Velocity Detection and Translational Motion Estimate

The focus of this section is the development of a model that integrates both zero-velocity detection and displacement estimates for periods of movement simultaneously. Thus, an LSTM-based stacked neural network is proposed. This neural network is presented in Figure 7.

The first part of the network is composed of an LSTM layer with 30 units and a densely connected layer to identify the timesteps with zero-velocity. This forces the model to extract this information from the data which can be regarded as a feature. After, the output of that layer is concatenated with the inertial data input. This way, the input of the following layers is composed of inertial data of each timestep and its labeling. The model is composed of two extra LSTM with 30 units each. Afterwards, a dropout layer with a dropout rate of 0.25 was used to avoid overfitting. The final output (3D position) is calculated with a densely connected layer that reduces the output to 3D.

In the methodologies presented in Section 3.4, the inputs of the models are the inertial measurements from periods classified as periods of motion. In this method, distinctively from the last ones, the inputs used are composed by the inertial data of an entire trial. This input contains inertial data from periods of movement and of zero-velocity. The outputs include the displacement in each axis from the beginning of the trial. This displacement is calculated by subtracting the 3D ground truth position at the beginning of the trial from the 3D ground truth position in each timestep.

This stacked neural network was built to identify moving/stopped moments and estimate the displacement that occurs during a data collection trial. To achieve this result, the loss function of the neural network was customized so that, during the training process, it took into consideration, simultaneously, the classification error and the regression error.

This way, the loss function (*L*) for a trial with *N* samples can be defined as:(8)$$\begin{array}{c} L\left(y_{cl},{\hat{y}}_{cl},y_{reg},{\hat{y}}_{reg}\right)=\left(1-\alpha \right)\times L_{Binary}\left(y_{cl},{\hat{y}}_{cl}\right)+\alpha \times L_{MSE}\left(y_{reg},{\hat{y}}_{reg}\right)=\\ \left(1-\alpha \right)\times \left(-\frac{1}{N}\sum_{i=0}^{N-1}y_{cl}\times log\left(p\left({\hat{y}}_{cl}\right)\right)+\left(1-y_{cl}\right)\times log\left(1-p\left({\hat{y}}_{cl}\right)\right)\right)+\\ +\alpha \times \left(\frac{1}{N}\sum_{i=0}^{N-1}{\left(y_{reg}-{\hat{y}}_{reg}\right)}^{2}\right) \end{array}$$
in which $y_{cl}$ represents the ground truth zero-velocity labeling and ${\hat{y}}_{cl}$ is the respective model estimate. $p({\hat{y}}_{cl})$ represents the probability of the IMU having velocity zero in that timestep. $y_{reg}$ symbolizes the translation’s ground truth and ${\hat{y}}_{reg}$ is the model’s displacement estimate. α represents a manually set weight: that allows adjusting the importance of the classification and regression steps.

This model was trained using an Adam optimizer [36] with a learning rate of $5\times 10^{-3}$, weight decay of $1\times 10^{-5}$, and batch size 1. The same split as described in Figure 2 was applied. From the ten patients present in the train set, data from seven were used for training, and the data from the other three were used for validation. Early stopping was conducted to stop the training when the validation loss was not decreasing throughout 20 epochs.

## 4. Results

### 4.1. Zero-Velocity Detection

#### 4.1.1. Automatic Labeling of the Qualisys Data

The threshold *thresh* was defined as 1 cm, and the window size *N* as 30 samples since these were the values that optimized the results of the automatic labeling. This means that in each window of 30 samples—which corresponds to a window of 0.25 s considering the 120 Hz sampling rate—if the movement is above 1 cm, then it is considered that the body segment is moving. In other words, if the average velocity in that window is above 0.04 m/s, then it is considered that the body segment is moving. To introduce hysteresis, the value *T* was defined as 3. This implies that, if the body segment was previously at rest, then the threshold is elevated to 3 cm.

The results of the comparison of automatic labeling with manual labeling are presented in Table 1. The positive class was considered to be the “*stopped*” state.

First, it is essential to consider that manually identifying the stationary and movement periods using the position signal is a difficult task. Secondly, it is important to remind that this manual segmentation only acts as a term of comparison and that the automatic labeling algorithm allows identifying the periods in which the change in position is smaller than a predefined threshold. This allows accurately stating that, if the velocity of the movement in a period is small, then one considers that the IMU and associated body segment are stopped.

Overall, the results suggest that the developed algorithm is an adequate method to perform the labeling of the 3D placements, and that the generated labels can, therefore, be used as ground truth for the subsequently developed supervised machine learning models.

#### 4.1.2. Zero-Velocity Detectors

The accuracy of the different developed classifiers for performing zero-velocity detection is presented in Figure 8.

Analyzing the results, it is possible to conclude that all the chosen models can be successful in differentiating moments of movement from stationary moments.

Comparing the fixed threshold methods with the ML-based methods, it is possible to observe that the ML methodologies were capable of obtaining a slightly higher accuracy in the segmentation of zero-velocity periods (except the logistic regression model).

The application of a median filter as a post-processing step on the ML methods also slightly improves the accuracy of all the models. Analyzing the test set results, only the Logistic Regression model has an accuracy lower than 0.94. The rest of the models have a very similar performance.

This level of accuracy must be studied considering the level of error that exists in the production of the ground truth. These results look good enough considering the error intrinsic to the annotation. Additionally, we believe that, considering the final goal of this paper (which is to improve the translational motion estimate), this amount of error is not relevant in the calculation of the displacement considering that the data from the periods in which the errors occur is associated with a small displacement.

The model with the best accuracy in the test set was the Random Forest model with median filtering. An example of this classification on a trial is presented in Figure 9.

### 4.2. Translational Motion Estimates between Stationary Periods

To evaluate the translational motion estimates conducted using the different zero-velocity detectors and the different regressors, three different metrics were used to assess the displacement error in each axis: RMSE (root-mean-squared error), MAE (mean absolute error) and R${}^{2}$.

RMSE can be represented by:(9)$$RMSE=\sqrt{\frac{1}{N}\sum_{i=0}^{N-1}{\left(p_{i}-\hat{p_{i}}\right)}^{2}}$$

To allow a more straightforward interpretation of the results, the MAE of the displacement in each axis was also evaluated:(10)$$MAE=\frac{1}{N}\sum_{i=0}^{N-1}\left|\left(p_{i}-\hat{p_{i}}\right)\right|$$

In both equations, $p_{i}$ represents the true position in a given axis in the timestep i and $\hat{p_{i}}$ is the estimated position. Since, at the beginning of each trial, the IMUs coordinate frame was reinitialized to match the Qualisys world frame, to compare $p_{i}$ and $\hat{p_{i}}$, no rotation needs to be applied. RMSE and MAE were calculated for the *x*-, *y*-, and *z*-axis. It was considered that the starting point of all the trials was the point of coordinates [0,0,0].

It is also important to understand the average prediction error of each model in the 3D trajectory. With that purpose, the absolute trajectory error (ATE) of the position was also assessed. The absolute trajectory error of the position can be regarded as a 3D RMSE. This way, ATE is defined as [39]:(11)$$ATE=\sqrt{\frac{1}{N}\sum_{i=0}^{N-1}{∥\Delta p_{i}∥}^{2}}$$
in which $∥\Delta p_{i}∥$ represents the Euclidean distance between the estimate and the ground truth:(12)$$∥\Delta p_{i}∥=d\left(p_{i},{\hat{p}}_{i}\right)=\sqrt{{\left(p_{\left(x,i\right)}-{\hat{p}}_{\left(x,i\right)}\right)}^{2}+{\left(p_{\left(y,i\right)}-{\hat{p}}_{\left(y,i\right)}\right)}^{2}+{\left(p_{\left(z,i\right)}-{\hat{p}}_{\left(z,i\right)}\right)}^{2}}$$

Additionally a 3D MAE was also calculated:(13)$$MAE\left(\Delta p_{i}\right)=\frac{1}{N}\sum_{i=0}^{N-1}∥\Delta p_{i}∥$$

The calculation of RMSE and MAE provides a single, easy to compare value. However, it is essential to mention that these metrics are time-sensitive: if a position estimate mistake happens at the beginning of a trial, then the error tends to be higher then if the same error takes place as the end of the trial [39]. Besides, it is also important to keep in mind that the RMSE gives higher weight to more significant errors [40]. All the metrics were evaluated individually for each trial and, therefore, the average values between the test trials are presented with the respective standard deviations.

Another interesting regression performance metric is R${}^{2}$, which allows assessing how close the predicted variable fits the ground truth. In this case, R${}^{2}$ can be calculated as [41]:(14)$$R^{2}=\frac{Var(\hat{p})}{Var(p)}=Correlation{\left(\hat{p},p\right)}^{2}$$

This index allows calculating the proportion of the variance in **p** that the developed model can explain. If the developed models predict perfectly the desired variable, then R${}^{2}$ is 1 [41].

The results of some of the different implemented methods are presented in Table 2. These include the results of the combination of different zero-velocity classifiers with different regressors. The results of the stacked neural network which performs both tasks (classification and regression simultaneously) are also presented.

Firstly, it is important to analyze the importance of the application of the zero-velocity updates. There is a significant improvement in applying the zero-velocity update using the different classifiers compared with the simple double integration of the signals without any correction. The use of drift correction also improves RMSE and MAE of all the trajectory estimates with a zero-velocity update. R${}^{2}$ also increases in the cases in which drift correction is applied.

The model which stopped moments identification leads to a trajectory estimate with a lower error and a higher R${}^{2}$ is the Six-Layer LSTM model combined with double integration and drift correction. In this case, the drift correction also improves this estimate on all axes. However, all the classifiers combined with double integration as regressor possess a similar performance between them before and after the drift correction.

Another interesting point is to note that the displacement estimates possess a lower average error in the z-axis. Provided that the IMUs are not always under high linear acceleration (which is true for human movement), then these are very accurate in terms of vertical measurements. In contrast, IMUs fail in horizontal measurements because there is no way to compensate the gyro drift around global Z (without proper magnetometer corrections). To remove gravity from the accelerometer readings, one needs to know the orientation of the sensor relative to Earth. Since both accelerometers and gyroscopes are affected by errors, the estimation of that orientation is never totally accurate. This inaccurate orientation estimation—that in this case was previously conducted by the Xsens MVN Link system—leads to errors during the gravity removal process which will affect less the measurements in the vertical axis than on the non-vertical axes. Besides, it is also important to consider that less movement occurs in the z-axis (the subjects major movements occur in the horizontal plane) and thus the removal of the gravity may also be optimized in axes in which less motion occurs.

To compare the different regressors, the same 6-LSTM classifier was used to conduct zero-velocity detection. Afterwards, the different regression models were used to estimate the displacement between zero-velocity periods. Starting by comparing the double integration with drift correction results with the ML regressors, one can conclude that these could not improve the overall displacement estimate during an entire trial.

By comparing the different regressors with each other, it is possible to observe that the “pre-trained” model achieves a lower average error on the x-axis while the “no initialization” model achieves a better performance on the y-axis. On the z-axis and for the 3D displacement estimate, the performance of these models is similar.

The stacked neural network methodology was experimented using three different α values for the loss function (Equation (Equation 8))—0.5, 0.8 and 1. A α value of 0.5 gives similar weight to the classification loss and to the regression loss. A α value of 0.8 gives higher weight to the regression loss. A α value of 1 means that the loss function is equivalent to the mean squared error loss function.

By comparing the stacked neural networks with different α in the loss function, it is possible to see that the results are similar for the three methods. The network in which α was set to 1 achieves better results on the x-axis average error than the other ones, which results in a lower error in the 3D displacement estimate.

The stacked neural networks method allows to train networks that are capable of simultaneously detecting zero-velocity periods and translation estimate. This way, it is probable that, during the training of the neural network, the model is capable of understanding what are the detected stationary periods that will support an improved translation estimate. This way, some errors that could exist on the zero-velocity “ground truth” will be annulled by the other goal of the neural network that is to find a better translation estimate (in the cases in which α > 0.5, the translation estimate is the major objective of the neural network).

## 5. Conclusions

This work focused on providing an improvement in the position estimate using inertial data of different body segments (to which an IMU is attached) when a subject is performing several movements using inertial data. For this purpose, two objectives were defined: (1) zero-velocity detection for identifying periods in which the IMU is stationary; and (2) regression of translations that took place in periods of movement. Lastly, an ML model that was capable of accomplishing both tasks (zero-velocity detection and displacement estimate) was also evaluated.

The developed zero-velocity detectors provided an apparent accurate detection of IMU stationary periods which seems to have allowed constraining the error that accumulates during the integration of the noisy inertial measurements. The application of these detectors to correct the velocity estimation allowed an enhanced tracking when compared with the simple double integration of the accelerometer data. It is important to state that the improvement of this detection, as well as an enhanced definition of the ground truth, would allow a better tracking. The removal of the linearly estimated drift between zero-velocity moments provided an additional improvement.

Relatively to the regression models, used to estimate translations between zero-velocity moments, no significant improvements in comparison with more straightforward methods—as the double integration with drift removal—were achieved. The approach that combined both zero-velocity detection and translations estimate showed promising results.

The majority of the work existent in this area is dedicated to different variants of the problem tackled in this project. Some authors use inertial data to track subjects walking and estimate the trajectory of their path or their velocity. Regarding zero-velocity detection, the existent work concentrates into zero-velocity detection during gait using IMUs placed on the foot, lacking both movement diversity and data from several body segments. Others concentrate on evaluating 6-DoF pose estimate for tracking in areas as robotics and automation and, therefore, do not focus on the more complex task of tracking human motion. Besides, to the best of our knowledge, there is not any type of similar work developed using the same dataset which constitutes an additional difficulty for any kind of comparison. This absence of work in the area made the comparison of the developed methods with prior techniques difficult.

This work provides several research opportunities. First, further optimization must be performed in all the developed models. Zero-velocity detection was performed with high accuracy, although it can be improved with more exploration and optimization. The regression estimate presented results with lower performances and, in this way, future work can be focused on this area. Some examples of work that can potentially be performed on this part following the methods implemented in this project include: initializing the weights of an LSTM-based model using simultaneously data from the three-axis to provide a warm start to the network and the ability to capture information at the same time from inertial data from different axes; conducting warm start for the stacked neural network methodology—particularly on the first layer responsible for detecting stationary periods using the weights optimized during in the zero-velocity detection LSTM classifier. The work should also be focused in improving and solving the capacity to adapt to negative values presented by the approach that combined simultaneous zero-velocity detection and translations estimate as well as filtering the results obtained in that section.

Other future work includes the validation of the developed methods in different datasets. Although the used dataset presents wide motion variety and data from several body segments, it is vital to understand if the developed models can generalize to other parts of the body, different IMUs, and movements of different nature. The final goal should be to achieve a model that can improve human motion tracking being applied to data from any IMU that is placed in any body segment during the execution of any type of movement.

## Figures and Tables

**Figure 1 sensors-20-06383-f001:**
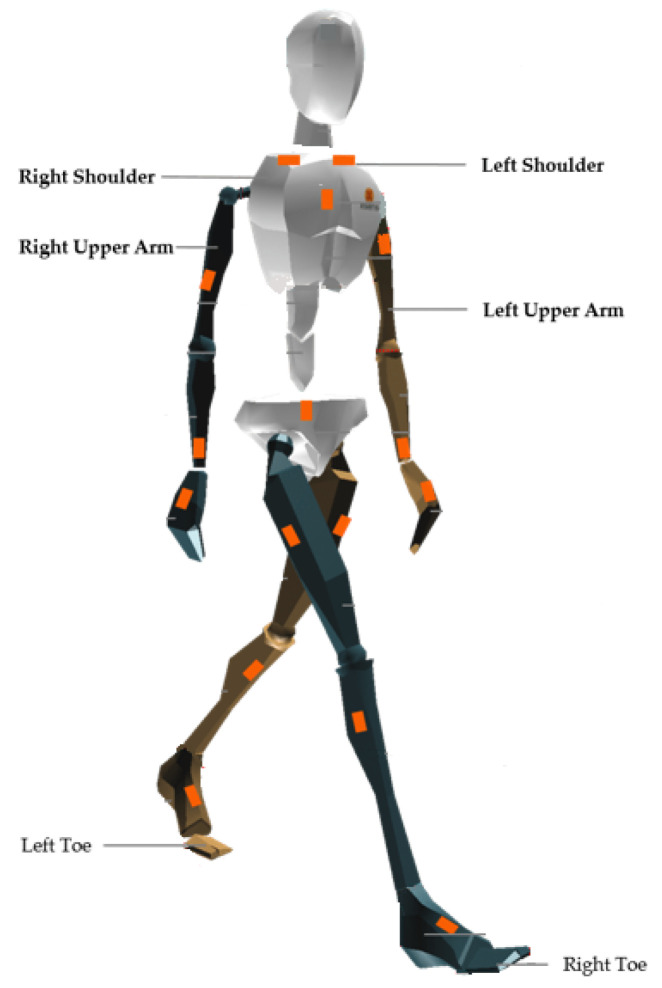
Xsens Avatar with indication of the six body placements in which both inertial data and ground truth 3D position are provided, extracted from [33].

**Figure 2 sensors-20-06383-f002:**
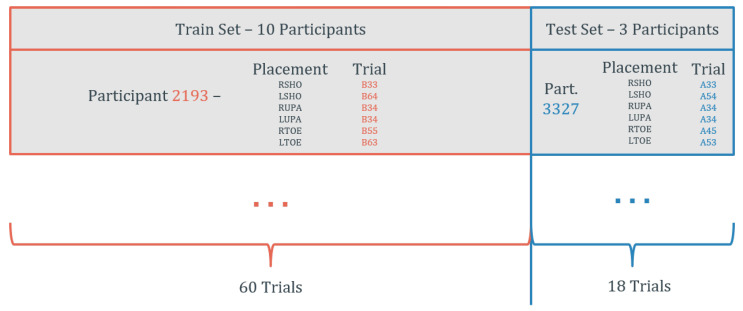
Train and test split (RSHO, Right Shoulder; LSHO, Left Shoulder; RUPA, Right Upper Arm; LUPA, Left Upper Arm; RTOE, Right Toe; LTOE, Left Toe).

**Figure 3 sensors-20-06383-f003:**
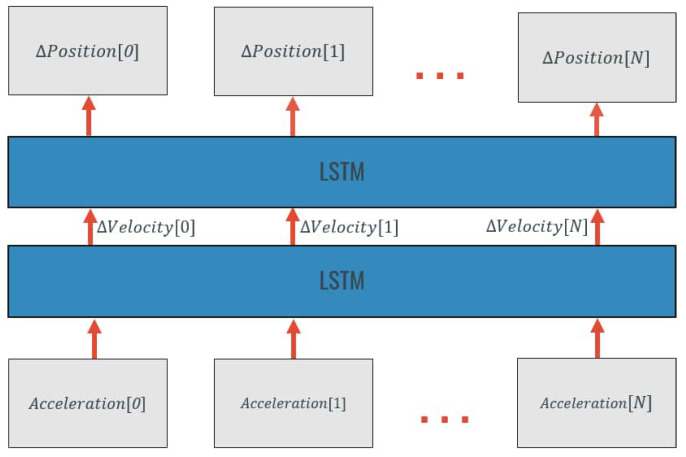
“Smart” double integration by a two-Layer LSTM (Acceleration[i], linear acceleration at instant i; ΔVelocity[i], velocity change at instant i; ΔPosition, displacement at instant i; LSTM, Long-Short-Term Memory Network Layer).

**Figure 4 sensors-20-06383-f004:**
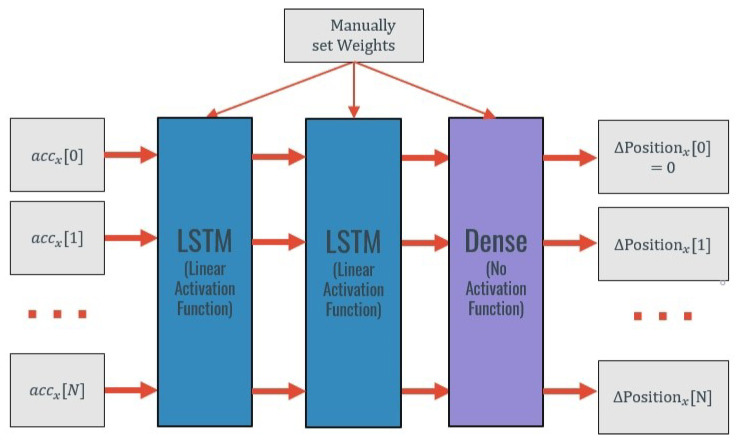
Representation of the warm-Start with “integrative” weights methodology—example for the x-axis.

**Figure 5 sensors-20-06383-f005:**
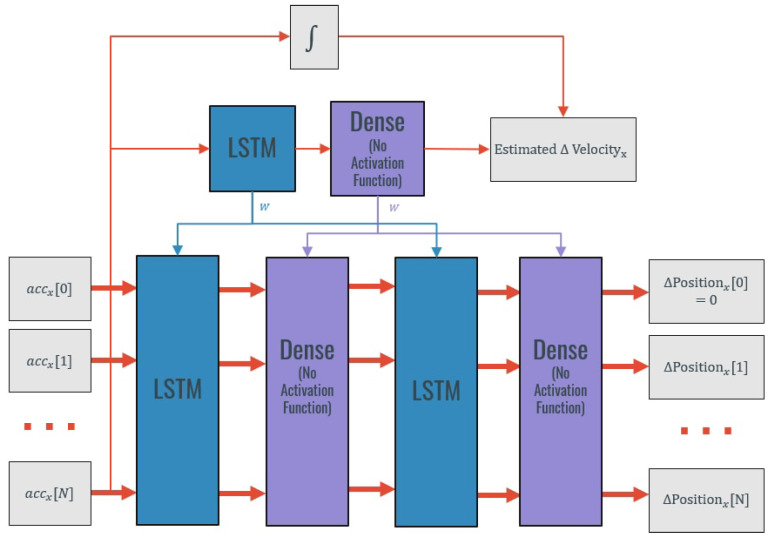
Representation of the warm-Start with “pre-trained” weights methodology—example for the x-axis.

**Figure 6 sensors-20-06383-f006:**
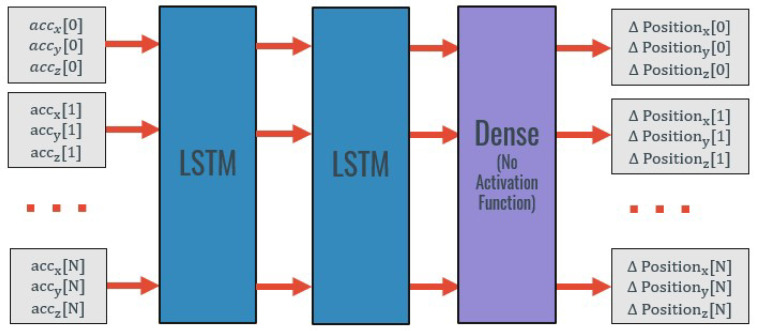
Neural network architecture with no type of warm start for three-axis displacement estimate.

**Figure 7 sensors-20-06383-f007:**
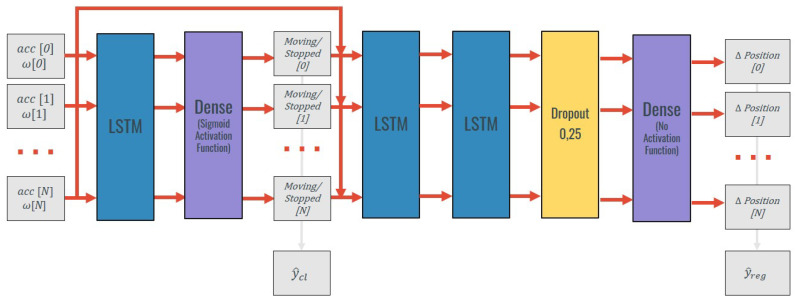
Stacked neural network for simultaneous classification and regression.

**Figure 8 sensors-20-06383-f008:**
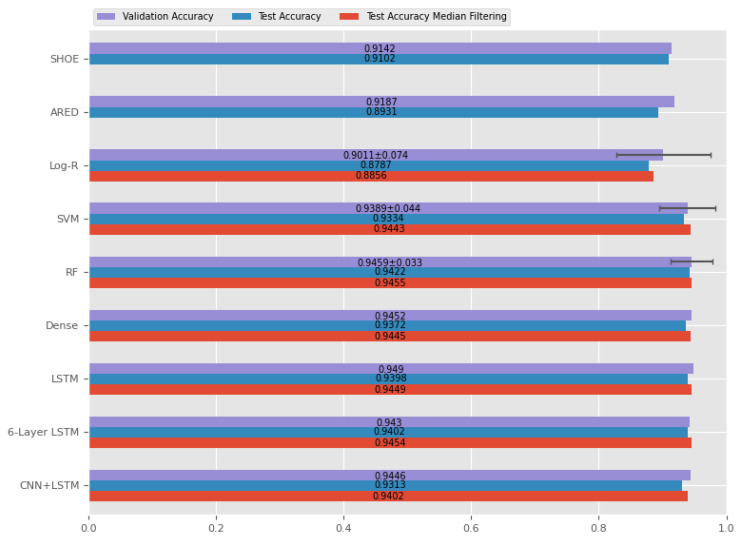
Accuracy of the different used classifiers (SHOE, Stance Hypothesis Optimal Estimation; ARED, Angular Rate Energy; Log-R, Logistic Regression; SVM, Support Vector Machine; RF, Random Forest; Dense, Densely Connected Neural Network; LSTM, Single Layer LSTM; CNN + LSTM, Convolutional Layers + LSTM Layers).

**Figure 9 sensors-20-06383-f009:**
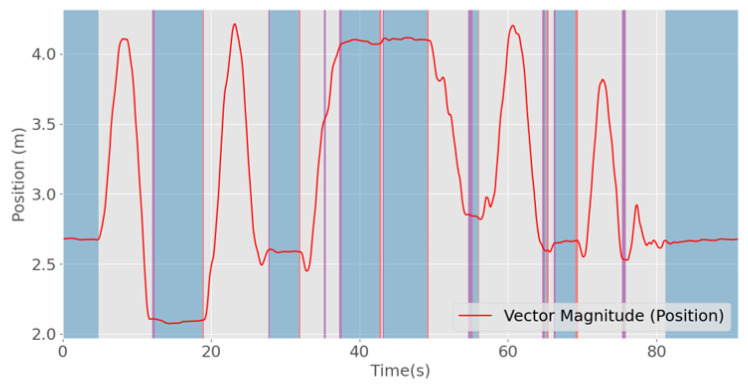
Example of a trial classification using the Random Forest classifier with and without median filtering (blue background, true positives (correctly identified stopped periods); gray background, true negatives (correctly identified periods of movement), purple background, false positives (wrongly labeled as stopped periods); red background, false negatives (wrongly labeled as moving periods)).

**Table 1 sensors-20-06383-t001:** Results of the automatic labeling when compared with the manual labeling (RSHO, Right Shoulder; LSHO, Left Shoulder; RUPA, Right Upper Arm; LUPA, Left Upper Arm; RTOE, Right Toe; LTOE, Left Toe).

Placement	Accuracy	Precision	Recall
RSHO	0.9564 ± 0.0036	0.9722 ± 0.0126	0.9471 ± 0.0126
LSHO	0.9375 ± 0.0187	0.9499 ± 0.0264	0.9345 ± 0.0264
RUPA	0.9134 ± 0.0590	0.9290 ± 0.0333	0.9107 ± 0.0333
LUPA	0.9469 ± 0.0151	0.9691 ± 0.0166	0.9359 ± 0.0166
RTOE	0.8907 ± 0.0270	0.9487 ± 0.0221	0.9126 ± 0.0221
LTOE	0.8881 ± 0.0358	0.9306 ± 0.0365	0.9268 ± 0.0365
Total	0.9222 ± 0.0415	0.9499 ± 0.0376	0.9279 ± 0.0290

**Table 2 sensors-20-06383-t002:** MAE, RMSE, and R${}^{2}$ of the displacement estimate in each axis of the full trial 3D displacement estimate using the different classification and regression models.

Zero-Velocity Classifier	Regressor	Error	p${}_{x}$	p${}_{y}$	p${}_{z}$	|| p||
None	Double Integration	RMSE (m)	28.71 ± 24.55	22.75 ± 22.03	6.60 ± 5.61	23.82 ± 16.35
		MAE (m)	22.45 ± 19.98	17.30 ± 17.25	5.25 ± 4.40	32.41 ± 22.80
		R${}^{2}$	0.15 ± 0.13	0.13 ± 0.15	0.06 ± 0.11	0.12 ± 0.08
Random Forest	Double Integration	RMSE (m)	1.41 ± 1.23	0.81 ± 0.62	0.69 ± 0.37	1.99 ± 1.10
		MAE (m)	1.25 ± 1.18	0.65 ± 0.45	0.57 ± 0.33	1.77 ± 1.0
		R${}^{2}$	0.64 ± 0.26	0.45 ± 0.26	0.20 ± 0.24	0.11 ± 0.14
Random Forest	Double Integration + Drift Correction	RMSE (m)	0.60 ± 0.39	0.56 ± 0.49	0.37 ± 0.23	1.02 ± 0.45
		MAE (m)	0.51 ± 0.37	0.47 ± 0.44	0.31 ± 0.20	0.92 ± 0.43
		R${}^{2}$	0.85 ± 0.13	0.63 ± 0.32	0.39 ± 0.31	0.35 ± 0.31
6-LSTM	Double Integration	RMSE (m)	1.25 ± 1.00	0.94 ± 0.78	0.68 ± 0.40	1.90 ± 1.04
		MAE (m)	1.08 ± 0.95	0.75 ± 0.65	0.57 ± 0.34	1.67 ± 0.98
		R${}^{2}$	0.62 ± 0.24	0.43 ± 0.32	0.25 ± 0.28	0.09 ± 0.11
6-LSTM	Double Integration + Drift Correction	RMSE (m)	0.50 ± 0.31	0.51 ± 0.49	0.42 ± 0.21	0.93 ± 0.44
		MAE (m)	0.42 ± 0.28	0.43 ± 0.43	0.35 ± 0.18	0.83 ± 0.39
		R${}^{2}$	0.86 ± 0.12	0.63 ± 0.30	0.31 ± 0.28	0.43 ± 0.13
6-LSTM	A—“Integrative”	RMSE (m)	1.63 ± 1.64	1.58 ± 2.37	1.01 ± 0.89	2.68 ± 2.85
		MAE (m)	1.44 ± 1.54	1.31 ± 2.20	0.85 ± 0.80	2.38 ± 2.65
		R${}^{2}$	0.57 ± 0.28	0.42 ± 0.31	0.21 ± 0.27	0.09 ± 0.10
6-LSTM	B—“Pre-trained” (w/drift removal)	RMSE (m)	0.87 ± 0.59	0.95 ± 0.84	0.31 ± 0.29	1.43 ± 0.91
		MAE (m)	0.71 ± 0.55	0.78 ± 0.72	0.24 ± 0.25	1.80 ± 1.29
		R${}^{2}$	0.68 ± 0.24	0.51 ± 0.32	0.38 ± 0.33	0.26 ± 0.30
6-LSTM	C—No initialization	RMSE (m)	1.29 ± 0.75	0.52 ± 0.2	0.35 ± 0.16	1.47 ± 0.72
		MAE (m)	1.09 ± 0.66	0.42 ± 0.19	0.29 ± 0.14	1.30 ± 0.65
		R${}^{2}$	0.62 ± 0.28	0.17 ± 0.16	0.09 ± 0.08	0.23 ± 0.24
Stacked Neural Network (α = 0.5)		RMSE (m)	0.81 ± 0.16	0.44 ± 0.06	0.19 ± 0.09	0.95 ± 0.15
		MAE (m)	0.62 ± 0.13	0.37 ± 0.07	0.15 ± 0.06	0.78 ± 0.14
		R${}^{2}$	0.57 ± 0.21	0.11 ± 0.10	0.12 ± 0.08	0.40 ± 0.22
Stacked Neural Network (α = 0.8)		RMSE (m)	0.78 ± 0.21	0.44 ± 0.06	0.19 ± 0.08	0.93 ± 0.21
		MAE (m)	0.59 ± 0.13	0.36 ± 0.06	0.15 ± 0.05	0.78 ± 0.14
		R${}^{2}$	0.50 ± 0.22	0.12 ± 0.13	0.19 ± 0.12	0.39 ± 0.26
Stacked Neural Network (α = 1)		RMSE (m)	0.54 ± 0.18	0.42 ± 0.07	0.20 ± 0.08	0.73 ± 0.17
		MAE (m)	0.44 ± 0.14	0.35 ± 0.07	0.15 ± 0.05	0.64 ± 0.14
		R${}^{2}$	0.68 ± 0.24	0.51 ± 0.32	0.38 ± 0.33	0.26 ± 0.30

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
