# Peer review of "Machine Learning Improvements to Human Motion Tracking with IMUs"

_sensors, 2020, doi:10.3390/s20216383_

Round 1
Reviewer 1 Report
This paper presents a work that calibrates IMU with an optical motion tracker for human motion position tracking using machine learning methods, addressing error accumulating problem when deriving positions from the IMU data using double integration. The problem the work addressing is fundamental to IMU sensors for position measurement, and thus would attract interest from broad readers. However, the paper is not publishable in the current form due to the results presented sound preliminary and could be improved.
The measurement errors presented in table 2 are huge, with the best of about 0.5 meter (px RMSE = 0.54, using Stacked Neural Network). I would suggest to compare the results to the state-of-the art in the literature if possible, and also justify how this accuracy is practical in real applications.
The accuracy of “ground truth” is only 92.22% (Table 1), which is not sufficient. I would suggest the authors to look into what is the cause of the almost 8% error in the ground truth data and further to improve the labelling.
For the only binary classification problem, the models of the zero-velocity moments detection have only about 94%. The author may also need to analyze error sources and justify whether this accuracy level is good enough. As this about 6% error will be carried on to position estimation.
The paper only present RMSE and MAE for the measurement of accuracy. However, both of them only reflect mean absolute error. Normally, R2 would be necessary for regression performance measurement, which would provide how close of the estimated variable fits the ground truth.
The paper needs to report how much zero-velocity detection is improved using ML methods, compared to non-ML method, e.g. threshold method (if possible). This would clarify why using machine learning to segment zero-velocity periods.
Does the paper suggests that it is not necessary to use regressor, as the double integration method companioning with the zero-velocity detection would achieve similar performance. Furthermore, why the combined method (simultaneous zero-velocity detection and position estimation) works better should be explained.
In the Related Work, lots of terms used that are not understandable by readers.
Some minor points.
Page 2 line 70, What is “position fixes”
Page 3 lines 90-94, references are needed.
Line 152 what is “E-glove”?
Line 153, what is the definition of a “trial”?
P5, it is not necessary to present the algorithm table. The description is enough.
P6, line 203 “In all of them it was used as label the binary label which indicates the state.” — it is hard to understand. Please rephrase.
Figure 2 - 6, description of the elements in the figures is needed for the captions of the figures.
Line 271 what is “Warm-Start”?
Line 290, “were manually initialized to a value close to zero” — I think it is randomly initialized to close to zero, according to the description of following sentences.
Author Response
The responses are answered in the word file in appendix

Reviewer 2 Report
The authors claim to propose an improved human motion tracking method by applied machine learning method. The experimental results were validated by the conventional optical system. However, the methodology of the machine learning that the author provided the experimental environment didn’t convince the reviewer well. The comments are as follows.
- Please rewrite the abstract since it didn’t reveal the importance of this research work.
- It is not so easy to follow the article context. There are many sentences with complicated structures and meanings. Some grammar errors happened.
- Abbreviations have to be defined when they are first used. Please justify those phrases in the abstract. Additionally, please define the meaning of the abbreviation of AI in line 77, even though we know it means the artificial intelligence. Also, so as the LSTM in line 87 and RIDI in line 100, and so on.
- The support vector machine (SVM) is a classification method of machine learning. I It is weird to describe this method using the description in line 86: “…applying two Support Vector Machines (SVMs).” Was the original meaning “two different kernels of SVMs” ?
- Since the technique proposed in the manuscript would not real-time recognize the human motions, why the authors concern the computational complexity in training procedures?
- It seems 6 different motions were adopted to verify the proposed model. But the number of each motion is missing. Please also visualize the locations of the IMUs that on a human body.
- Please explain the meanings of “participant 2193” and “part. 3327” in figure 1. And also please define the phrases: RSHO, LSHO,… that listed in Fig. 1.
- In order to convince the reviewer and the readers, please shows the raw data and the corresponding labeling results.
- Please explain why the authors need not to apply activation functions in some dense layers.
- In the table 1, the scores of precision motions are higher than that of accuracy. It seems that the model was over-fitting.
Author Response
The responses are in the word file in the appendix.

Round 2
Reviewer 1 Report
Thank the authors for answering the questions and the improvement to the manuscript. I still concern about the extent of the questions solved is sufficient to endorse a publication. The authors answered questions in the response document, but it looks there is no actions taken to explain/improve in the manuscript. For example, in terms of the overall low accuracies of the position tracked by the IMUs, I think normally readers would also have similar concerns as mine. It would be nice or at least have a discussion in the manuscript. However, I actually still concern about the feasibility of the presented methods with such high errors in practical use, although the authors have explained that they have difficulty in comparing with other similar works. I would not go through the rest responses in detail in this review and decide not to endorse it for publication for the current form.
Reviewer 2 Report
The authors improved the quality of this manuscript, but it can be better. They offered a nice method for human motion tracking and several method comparisons. Some questions are listed as follow.
In abstract section:
- Does the phrase “inertial recording” in line 3 mean signal recording of inertial motions?
- In line 3, does the sentence “The solution of this problem is fundamental to improve…” mean “The fundamental solution of this problem is to improve…” or “The solution of this fundamental problem is to improve…”?
- Please briefly list the machine learning methods to inform the readers your strategies.
- In line 12, please briefly explain the “double integration with drift removal for translation estimate.” Double integration of what? Since, as the authors mentioned, the translation estimate was an important part in this manuscript, please also well define the phrase “translation estimate.” Does it mean to estimate the translational motions of sensors?
- Please quantify the so called “higher performance” of your position tracking methods mentioned in line 14.
In the sections of introduction and related work
- Please avoid use the word “translation” to represent “translational motions.”
- In line 189, the sentence “Additionally, data from 189 a particular participant is only present in the training set or in the test set” is not a standard approach in modeling. The outlier should be removed from either the training set or testing set. It should not be used for modeling or to validated the proposed model.
- It is confusing in the section of 3.3.1 automatic labelling of the qualisys data. The authors mentioned that their dataset was labelled manually but the section title indicates the data was automatic labelling. Meanwhile, it is also confusing that whether the dataset of ground truth was offered from the so-called Qualisys motion system or labeled by authors’ algorithm. How to define the threshold in the algorithm? Automatically defined by the algorithm or manually defined by users? Since the threshold values is critical to define the states of zero-velocity, the authors have to clearly explain their methods.
- It should be typo in Eq. (1). It should have a square on the angular velocity term.
- Are the threshold values used in Eq. (1) and (2) different? Even though the authors define the threshold as 1 cm or 3 cm as mentioned respectively in line 454 and 460, the unit of threshold value was mismatched with that in Eq. (1) and (2).
- A missed item or phrase in line 460.
